# Effect of Cold Rolling and Annealing on the Microstructure and Texture of Erbium Metal

**DOI:** 10.3390/ma15041370

**Published:** 2022-02-12

**Authors:** Shiying Chen, Xiaowei Zhang, Zongan Li, Shuang Wang, Yixuan Wang, Jinying Li, Daogao Wu, Zhiqiang Wang, Dehong Chen, Wenli Lu, Ning Mao, Wensheng Yang, Minglei Xu

**Affiliations:** 1National Engineering Research Center for Rare Earth, GRINM Group Co., Ltd., Beijing 100088, China; chenshiying1023@163.com (S.C.); zonganli@126.com (Z.L.); Wangshuang_WJ@163.com (S.W.); wangyx8945@163.com (Y.W.); 13278885733@163.com (J.L.); daodao173@163.com (D.W.); wzq97122@126.com (Z.W.); chen-dh@126.com (D.C.); lonely1304178568@163.com (W.L.); maoning234@163.com (N.M.); yws3179608@163.com (W.Y.); xml783235495@163.com (M.X.); 2GRIREM Advanced Materials Co., Ltd., Beijing 100088, China; 3General Research Institute for Nonferrous Metals, Beijing 100088, China

**Keywords:** erbium target, cold rolling, annealing, microstructure, crystal orientation

## Abstract

Erbium metal with purity ≥ 99% was cold rolled to 30%, 40%, 50%, and 60% deformations and the Er metal of 60% deformation was annealed at different temperatures for 1 h. The effect of cold rolling deformation and annealing on the microstructure and texture evolution of Er metal was investigated by XRD, EBSD, Microhardness tester, and OM. P is the orientation index, which is used to judge the preferred orientation. The research results showed that grains were broken and refined gradually with increasing deformation, the average grain size was 3.37 µm, and the orientation distribution was uniform for 60% deformation; deformation twins appeared in the grain when the deformation was less than 40%, which contributed to the generation of (0001) plane orientation. Comparing with the initial state, the (01$\overset{-}{1}$0) plane orientation gradually weakened and the (11$\overset{-}{1}$0) plane orientation had a trend of further strengthening with the increasing deformation; the ($\overset{-}{1}$2$\overset{-}{1}$0) plane orientation remained unchanged, but there was a gradual weakening trend when the deformation was greater than 50%. For 60% deformation of Er metal, the deformed microstructure was replaced by fine equiaxed grains with the increasing annealing temperature, and the high-performance Er metal with fine and uniform equiaxed grains can be obtained under annealing at 740 °C for 1 h.

## 1. Introduction

In recent years, rare earth oxide films have received widespread attention due to their high dielectric constant, large band gap width, and good thermal stability [1]. Er_2_O_3_ film is an ideal candidate for high dielectric materials and has a strong competitiveness in the new generation of metal oxide semiconductor field effect transistor (MOSFET) gate dielectrics [2,3,4]. Moreover, Er_2_O_3_ thin film can be applied to optical devices, tritium permeation barrier (TPB), and hydrogen storage [5,6]. Furthermore, erbium (Er) thin film is an ideal neutron generator for tritium target material in nuclear industry [7,8,9]. Erbium target is the key raw material of erbium oxide film and erbium film, and its quality directly affects the functional characteristics of the films. Therefore, it is necessary to obtain high-quality sputtering Er target with high purity, fine grain, uniform structure, and preferred orientation.

However, as-cast Er metal can’t satisfy the specifications of industrial sputtering targets, the grains usually need to be refined and crystal orientation should be optimized by rolling and annealing. Most rare earth metals have hexagonal close-packed (HCP) structure and fewer slip systems; therefore, the plastic deformation is very difficult because the rolling deformation process can easily cause stress concentration and lead to cracking [10,11]. There is a lot of research involving magnesium [12,13,14] and magnesium alloy [15,16], and titanium [17,18] and titanium alloy [19,20,21], but there are few studies on the deformation of rare earth metal. Huang P. [22] studied the effects of annealing temperatures on the hardness and microstructure of high-purity metal scandium metal after hot forging, and obtained the optimum annealing process of 725 °C × 0.5 h. Wang S. [23] studied the effect of annealing temperature on the microstructure of high-purity Er target under 80% deformation after hot rolling and obtained the optimum annealing process of 570 °C × 1 h. However, hot deformation process can easily lead to target oxidation, serious loss of raw materials, and worse surface quality. Compared with hot deformation, cold deformation has the advantages of a controllable microstructure, less loss, and precision deformation. S. Taskaev [24,25,26,27] has studied the effects of cold rolling deformation and annealing temperature on the specific heat, magnetic and magnetocaloric properties of gadolinium, terbium, and dysprosium. However, the effect of cold rolling and annealing on the evolution of the microstructure and texture of Er metal with HCP structure has not been studied.

In the present study, the cold rolling and annealing of Er metal has been performed. The influence of cold rolling deformation and heat treatment temperature on the microstructure, texture, and hardness of the Er metal target were studied, and the deformation mechanism and optimum annealing process were obtained.

## 2. Experimental

The commercially pure Er metal (99 wt.%) samples with dimensions of 50 mm × 25 mm × 12 mm were cold rolled by total deformation of 30%, 40%, 50%, and 60%, with 0.3 mm reduction during each pass (when the deformation exceeds 60%, Er samples started to be fractured), and the stress relief annealing was at 520 °C for 1 h when the deformation was increased by 10%. The rolling was performed at a speed of 150 r/min by means of a two-high mill rolling machine (Wuxi Xingxiang Metallurgical Machinery Factory, Jiangsu, China) with a roll diameter of 500 mm. The 60% cold-rolled Er samples were vacuum annealed at 460 °C, 500 °C, 540 °C, 620 °C, 660 °C, 700 °C, 740 °C, 780 °C, and 820 °C for 1 h in a tube furnace with the vacuum degree less than 10^−1^ Pa (The annealing temperature was chosen according to the recrystallize(ation starting temperature of 460 °C~520 °C approximately), then the samples were cooled down naturally with the furnace. The schematic diagram of the cold rolling process was shown in Figure 1.

The samples were surface-grinded with silicon carbide sandpaper from 200 mesh to 5000 mesh and mechanically polished with 0.5 µm diamond spray polishing agent, samples for EBSD measurements need to be further polished with argon ion. The microstructure, grain orientation, grain boundary distribution, and grain size on the rolling direction (RD) and normal direction (ND) plane at the mid-width of the sample were observed by the MM-6 optical microscope (OM, produced by Shanghai Optical Instrument Factory No.6, Shanghai, China) and Electron Backscattered Diffraction (EBSD). EBSD analysis was carried out on using ZEISS-Sigma500 scanning electron microscope equipped (Zeiss Germany, Jena, Germany) with an Oxford-symmetry accessory. The grain orientation on the RD–ND plane was analyzed by Philips PANalytical X-ray Diffraction (XRD) unit (PW3373, Almelo, The Netherlands) with Cu-Kα radiation. The hardness (HV0.3) was tested by an HVS-1000A Vickers microhardness tester(Huayin Test Instrument Co. LTD, Laizhou, China), which indirectly reflected the recrystallization during the annealing process; the load of the microhardness tester was 3 N and the load time was 10s. The RD-ND surface of Er samples was shown in Figure 2.

## 3. Results and Discussion

### 3.1. Starting Material of Er Metal

Figure 3 shows the microstructure, inverse pole figure (IPF) map, and grain size distribution of the initial as-cast Er metal. In the Figure 3a, the grains of initial as-cast metal are mainly irregular equiaxed crystals and the grain size distribution is not uniform; some fine crystal grains are observed around the coarse equiaxed crystals, as shown in the circles of Figure 3a. Rare earth metal is prone to oxidation during polishing and corrosion treatment, therefore, there are a large number of black spots at the grain boundaries and inside the grains, which is identified by red arrows in Figure 3a. Figure 3b,c are the IPF map and grain size distribution tested by EBSD on the RD-ND plane, respectively, which show that the grain size distribution in the initial Er samples is not uniform, the size of large grains is larger than 200 µm, and that it is smaller than 50 µm for some fine grains. Meanwhile, the grain distribution is disordered and many fine grains are observed in the large grains. It can be observed from Figure 3b that the crystal orientation of the initial as-cast Er metal mainly includes ($\overset{-}{1}$2$\overset{-}{1}$0) and (01$\overset{-}{1}$0) plane on the RD-ND plane.

### 3.2. The Influence of Cold Rolling Deformation on the Microstructure and Texture of Er Metal

Figure 4 shows the IPF map of cold rolling Er samples. In Figure 4a, when the deformation comes to 30%, compared with the initial as-cast state, most of the grains are broken and refined, some twins are formed inside the grains (as identified by the yellow arrows), but there are lots of large deformed grains; furthermore, the crystal orientations mainly include ($\overset{-}{1}$2$\overset{-}{1}$0) and (01$\overset{-}{1}$0). When the deformation reaches to 40%, as shown in Figure 3b, massive structure disappears and the grains are further refined, there is also a small part of twins formed (as identified by the yellow arrows), and the crystal orientations mainly include ($\overset{-}{1}$2$\overset{-}{1}$0), (01$\overset{-}{1}$0), and (0001), which indicates that the formation of twins contributes to the (0001) orientation. When the deformation increases to 50%, the grains are a bit further refined, twins disappear, and the crystal orientation distribution is uniform; the reason for the twins’ disappearance is that the slip is activated under the heat conditions in which the deformation temperature increases, which is an important role in the release of stress concentration in plastic deformation and becomes the main mechanism for plastic deformation in the larger cold rolling deformation process. The result indicates that the deformation mechanism is twinning and slip synergistic at the beginning stage of cold rolling and then transforms to slip-based at large than 40% cold rolling deformation. The similar transformation of deformation mechanism has been observed in cold rolled titanium by Ren Y. et al. [28]. When the deformation increases to 60%, the grains are completely destroyed and dominated by fine deformed structures, the grain boundaries increase correspondingly, there are no new twins, and the crystal orientation distribution is more uniform, but some small crystal nuclei are formed inside the grain, as shown by the circle in Figure 4d.

Figure 5 is the relationship curve between deformation and average grain size. The average grain size of Er decreases from 6.87 µm at 30% deformation to 3.87 µm at 40% deformation rapidly. When the deformation is greater than 40%, the average grain size slowly decreases from 3.87 µm at 40% deformation to 3.37 µm at 60% deformation, combined with the fine microstructure in Figure 4d, which indicates that grains are almost broken and refined completely after 60% cold rolling deformation.

Figure 6 and Table 1 show the distribution of the grain characteristics of the Er after cold rolled deformations. It is observed that the deformation area accounts for 88.28%, and the recrystallized grains area accounts for 1.79% when the deformation is 30%. After rolling at 40% deformation, the deformation area increases to 96.21%. Moreover, when the deformation increases to 50%, the deformation area is 94.60%, which is approximately the same as that of 40% deformation. However, when rolling deformation increases to 60%, the deformation area drops to 85.09% and the recrystallized area increases to 6.60%. Therefore, it is concluded that the degree of deformation first increases, then stabilizes, and then gradually decreases, and the degree of deformation gradually decreases because the freshly recrystallized grains replace the original deformed microstructure, which is consistent with the results in Figure 4.

Figure 7 is the misorientation distribution of Er metal. It is observed that most of the misorientation angles are high-angle grain boundaries (θ > 15°), and there are few low-angle grain boundaries (0° < θ < 15°) in the initial as-cast metal. The high-angle grain boundaries gradually decrease and the low-angle grain boundaries gradually increase with deformation increasing. After 50% cold-rolled deformation, the content of low-angle grain boundaries increases to 83.25%. When the deformation increases to 60%, the low-angle grain boundaries reduce to 79.34%. A greater low-angle grain boundary content reflects that there will be greater degree of deformation and dislocation density [29,30]. Therefore, the results show that the increase trend of deformation degree and dislocation density is stable after 40% cold-rolled deformation. When the deformation reaches to 60%, the low-angle grain boundaries decrease, which indicates that partial recovery and recrystallization have occurred. This is consistent with the results of Figure 4 and Figure 6.

In order to further analyze the texture evolution in the cold rolling process, the crystal orientation on RD-ND plane of Er metal is analyzed by XRD. The results are shown in Figure 8, in which it can be clearly observed that the intensity of the (01$\overset{-}{1}$0), ($\overset{-}{1}$2$\overset{-}{1}$0), and (11$\overset{-}{1}$0) change with the increasing deformation. However, it can’t directly reflect the change trend of the crystal orientation relative to the initial as-cast state. Therefore, the orientation index *P* is introduced, calculated as following:(1)$$P=\frac{R}{R_{0}}$$
where *R* is intensity ratio of the two characteristic peaks of the sample, and *R*_0_ is intensity ratio of the two characteristic peaks of the sample with no preferred orientation, which is used as a criterion for orientation. *R* can be expressed as Equation (2):(2)$$R=\frac{I_{\left(H_{1}K_{1}L_{1}\right)}}{I_{\left({H{}^{\prime }}_{1}{K{}^{\prime }}_{1}{L{}^{\prime }}_{1}\right)}}$$
where $I_{\left(H_{1}K_{1}L_{1}\right)}$ and $I_{\left({H{}^{\prime }}_{1}{K{}^{\prime }}_{1}{L{}^{\prime }}_{1}\right)}$ are the diffraction intensity of the (H_1_K_1_L_1_) and (H‘_1_K′_1_L′_1_), respectively, with preferred orientation. *R_0_* is calculated as following:(3)$$R_{0}=\frac{I_{\left(H_{0}K_{0}L_{0}\right)}}{I_{\left({H{}^{\prime }}_{0}{K{}^{\prime }}_{0}{L{}^{\prime }}_{0}\right)}}$$
where $I_{\left(H_{0}K_{0}L_{0}\right)}$ and $I_{\left({H{}^{\prime }}_{0}{K{}^{\prime }}_{0}{L{}^{\prime }}_{0}\right)}$ are the diffraction intensity of the (H_0_K_0_L_0_) and (H′_0_K′_0_L′_0_), respectively, without preferred orientation. When the orientation index P equals 1, it indicates that there is no preferred orientation, when p deviates from 1, it indicates that the preferred orientation occurs. The greater the deviation, the more obvious the preferred orientation [31,32,33].

In the present study, I_0_ is the (11$\overset{-}{3}$2) orientation intensity value in each state, which is selected as the reference for all samples. The calculation results are shown in Figure 9 and Table 2, P(01$\overset{-}{1}$0), P(0001), P($\overset{-}{1}$2$\overset{-}{1}$0), and P(11$\overset{-}{1}$0) are the orientation index of (01$\overset{-}{1}$0), (0001), ($\overset{-}{1}$2$\overset{-}{1}$0) and (11$\overset{-}{1}$0), respectively. It is observed that the value of P(01$\overset{-}{1}$0) decreases from 1.791 at 30% deformation to 1.184 at 60% deformation, which reveals that the crystal orientation of (01$\overset{-}{1}$0) is gradually weakened with the increasing deformation. The value of P(0001) is greater than 1.2 when the deformation is 40% and is constant around 1 on other deformation conditions; this indicates that (0001) plane has a stronger orientation at 40% deformation, but there is no obvious preference under other cold rolling deformation. The value of P($\overset{-}{1}$2$\overset{-}{1}$0) is around 1 before 40% deformation, the deviation from 1 is large when the deformation is greater than 50%, but the orientation index is less than 1 at this time, which indicates that the preference orientation remains unchanged in the beginning deformation stage, and then presents a gradual weakened trend after 50% deformation. The value of P(11$\overset{-}{1}$0) is 1.552 at 30% deformation and increases to 1.771 at 60% cold rolling deformation, this indicates that the preferred orientation of the (11$\overset{-}{1}$0) plane is obvious and there is a further enhancement trend with the deformation increases.

### 3.3. The Effect of Annealing Temperature on Microstructure and Microhardness of Er Metal

When the deformation increases to 60%, the sample is dominated mainly by fine deformed structure; therefore, the Er metal with 60% deformation is selected to be annealed at 460 °C–820 °C for 1 h, and the microstructure after annealing treatment is shown in Figure 10. When the heat treatment temperature is 460 °C, a small amount of fine recrystallized crystal nuclei is formed, and the grain size distribution is nonuniform, but there is still a lot of deformed structure as shown in Figure 10a. When the temperature increases to 500 °C, there is obvious recrystallization and a large number of small, equiaxed grains. When the annealing temperature increases from 540 °C to 700 °C, the recrystallized grains grow rapidly, and the deformed structure disappears gradually so that the number of equiaxed crystals increase continuously and the uniformity of grain distribution improves constantly, but there are still some fine and uneven recrystallized grains. In Figure 10g, the deformed grains have been completely replaced by the uniform and fine equiaxed recrystallized grains, which indicates that the recrystallization process has finished when the temperature comes to 740 °C. When the temperature further increases to 780 °C and 820 °C, Figure 10h,i show that the grain boundaries migrate, the grains grow rapidly; however, some grains grow abnormally (as shown in the circle), so the uniformity of the crystal grain distribution is reduced.

The microhardness and average grain size of Er metal with 60% deformation after annealing for 1 h are plotted in Figure 11 and Table 3. When the annealing temperature increases from 460 °C to 540 °C, the grain of Er metal grows slowly, and the hardness decreases from 134.47 HV to 98.03 HV rapidly. This is due to the dislocation density decreases and the stored energy of microstructure decreases during the recovery and recrystallization process [34,35]. As the annealing temperature increases above 540 °C, the hardness tends to be steady and the average grain size rapidly increases from 5.43 µm at 540 °C to 21.52 µm at 700 °C, which is due to the internal stress disappearing and the energy of annealing completely transforms into grain growth. When the temperature reaches to 740 °C, the hardness is maintained at 97 HV and the crystal grain is 24.53 µm; then the grain size and hardness tend to be flat, which indicates that the recrystallization is finished at 740 °C. The grain growth trend is the same as that of the rolled copper target [36]. Moreover, the results of hardness and grain size changes at different annealing temperatures are consistent with the results of microstructure changes in Figure 10. It is concluded that when the deformation is 60%, a high-quality Er target with fine and uniform grains can be obtained after annealing at 740 °C × 1 h.

## 4. Conclusions

In this study, the effect of cold rolling and annealing temperature on grain size, grain uniformity, and texture of Er metal were investigated. The main experimental conclusions were summarized as follows:

(1) The grains of Er metal are gradually broken and refined with increasing deformation. When the deformation reaches to 60%, the grains are dominated by fine deformed structures, the average grain size is 3.37 µm, and the crystal orientation distribution is uniform;

(2) The deformation mechanism is transformed from the synergistic way of twinning and slip to the slip-dominated deformation at greater than 40% deformation and the generation of twins contributes to the formation of (0001) orientation;

(3) Compared with the initial as-cast state, the crystal orientation of (01$\overset{-}{1}$0) is weakened gradually with increasing deformation; the crystal orientation of ($\overset{-}{1}$2$\overset{-}{1}$0) remains unchanged in the beginning deformation stage, then gradually become weakened as the deformation is increased beyond 50%. The preferred orientation of the (11$\overset{-}{1}$0) is obvious, and there is a tendency to further strengthen;

(4) For the 60% cold rolling deformation of Er metal, the deformed microstructure was replaced by fine equiaxed grains as the annealing temperature increased. The average grain size is 24.53 µm and the grain size distribution is uniform after the optimal annealing process of 740 °C × 1 h.

## Figures and Tables

**Figure 1 materials-15-01370-f001:**
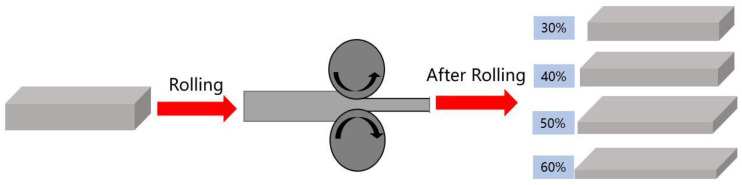
Schematic diagram of cold rolling process of Er metal.

**Figure 2 materials-15-01370-f002:**
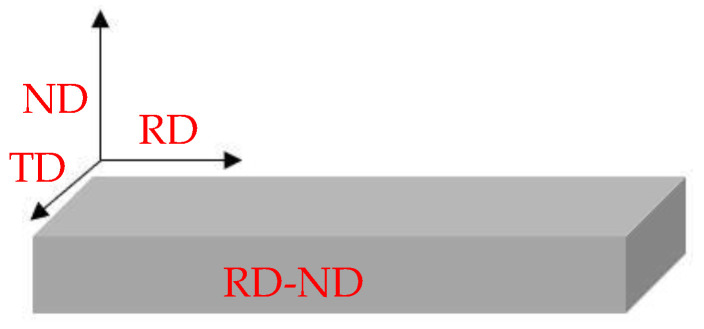
RD-ND surface of Er samples.

**Figure 3 materials-15-01370-f003:**
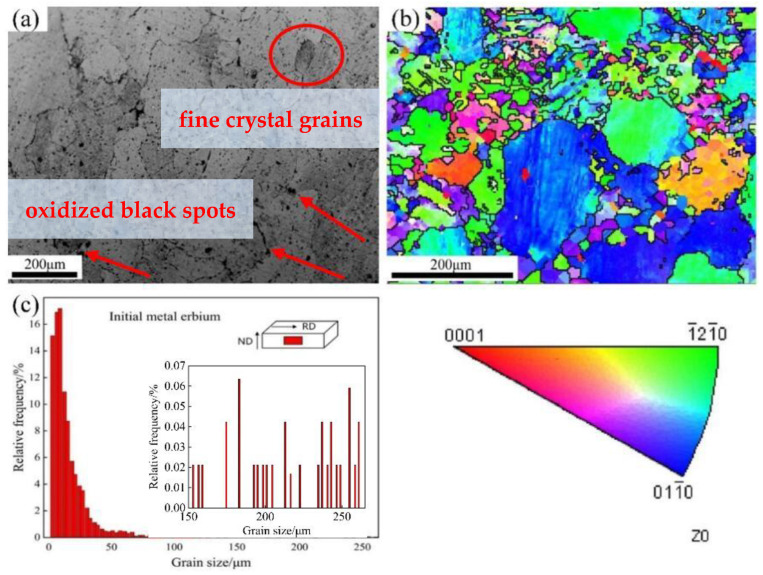
Microstructure, inverse pole figure (IPF) map and grain distribution of the initial as-cast metal: (**a**) microstructure; (**b**) crystal orientation; and (**c**) grain size distribution.

**Figure 4 materials-15-01370-f004:**
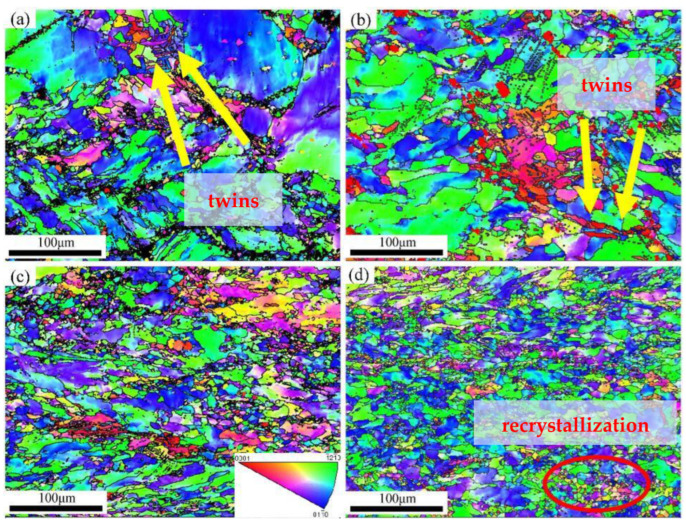
EBSD IPF maps during different cold rolling deformation: (**a**) 30%, (**b**) 40%, (**c**) 50%, and (**d**) 60%.

**Figure 5 materials-15-01370-f005:**
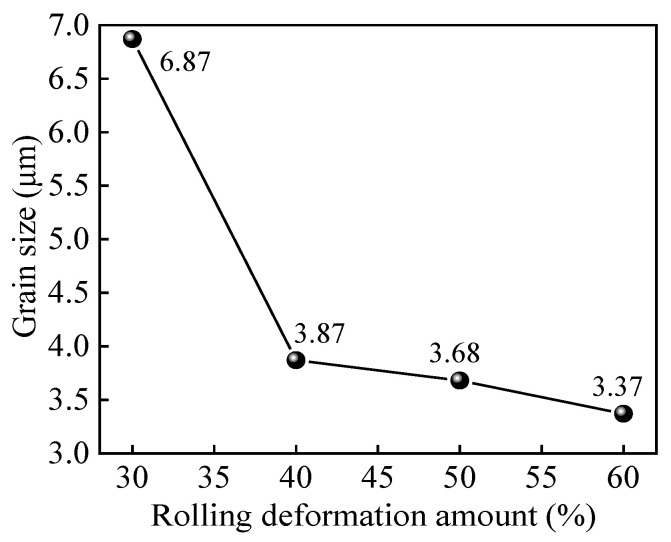
The relationship curve between deformation and average grain size.

**Figure 6 materials-15-01370-f006:**
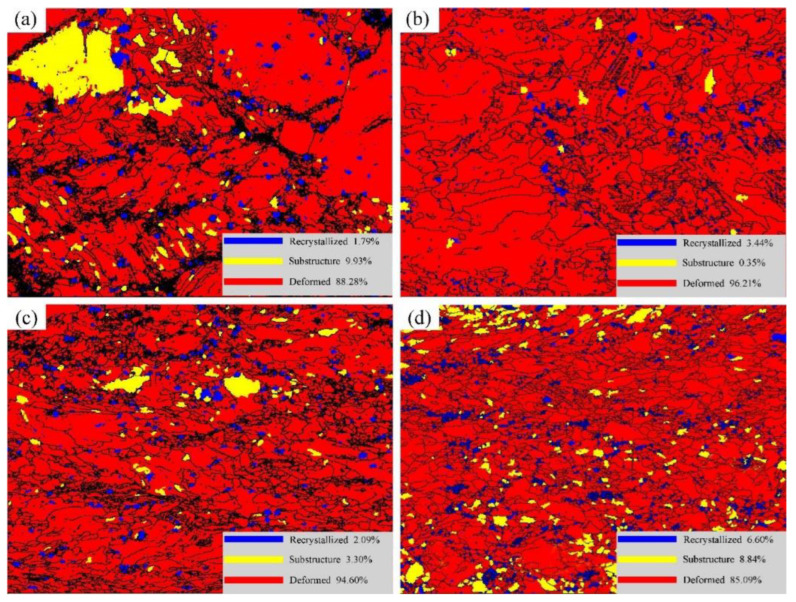
Grain distribution characteristics of Er metal with various cold rolling deformations: (**a**) 30%, (**b**) 40%, (**c**) 50%, and (**d**) 60%.

**Figure 7 materials-15-01370-f007:**
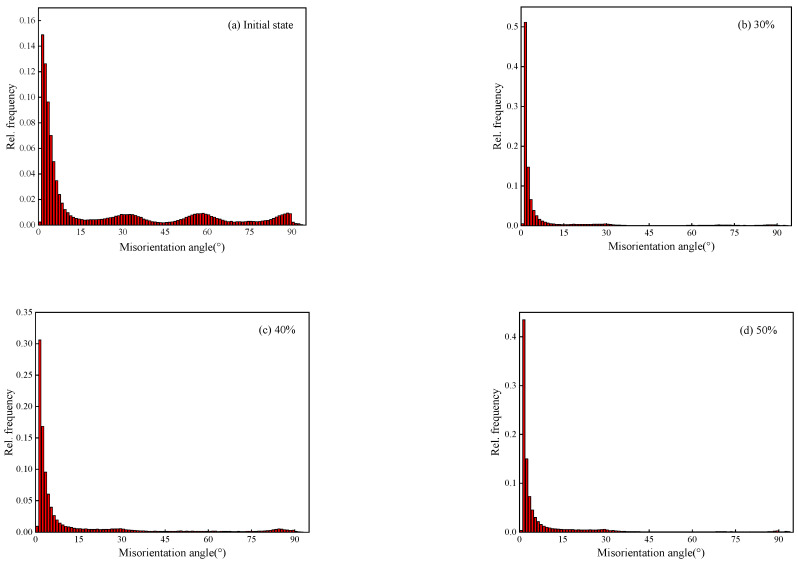
Misorientation distribution of Er metal with various cold rolling deformation: (**a**) initial state, (**b**) 30%, (**c**) 40%, (**d**) 50%, and (**e**) 60%.

**Figure 8 materials-15-01370-f008:**
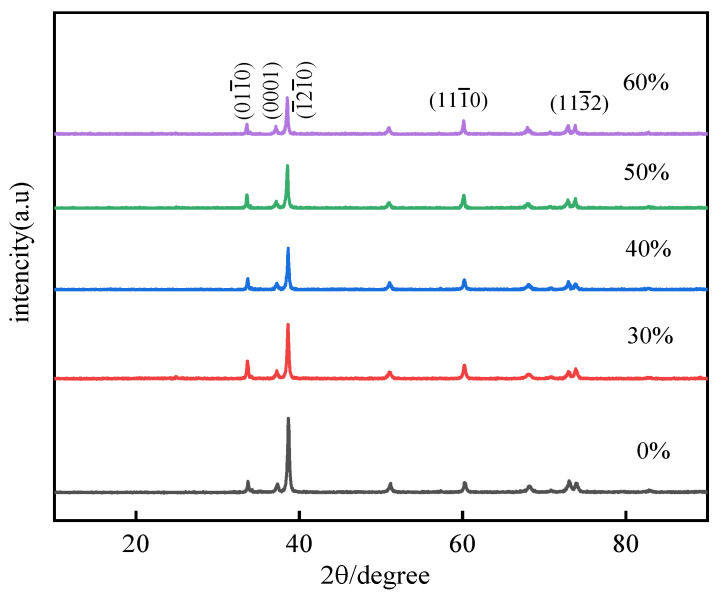
XRD map of Er metal with various cold rolled deformation.

**Figure 9 materials-15-01370-f009:**
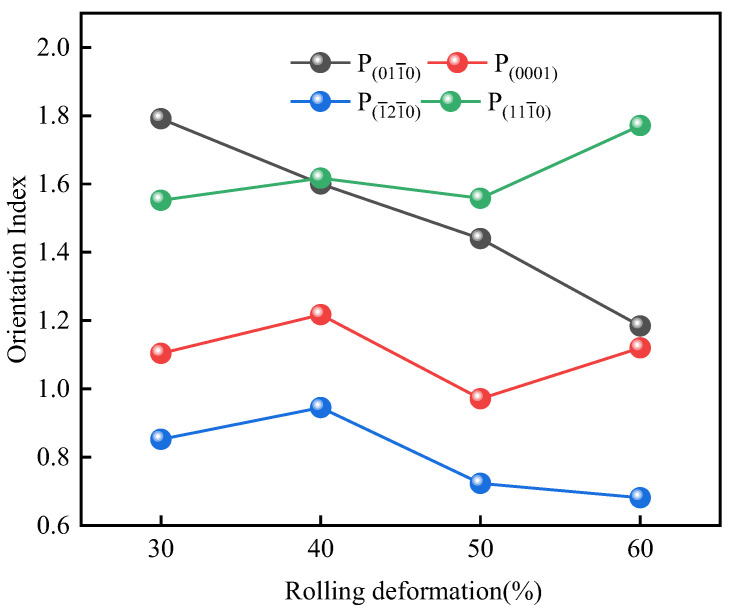
The relationship between the orientation index changes of Er metal with various cold rolling deformation.

**Figure 10 materials-15-01370-f010:**
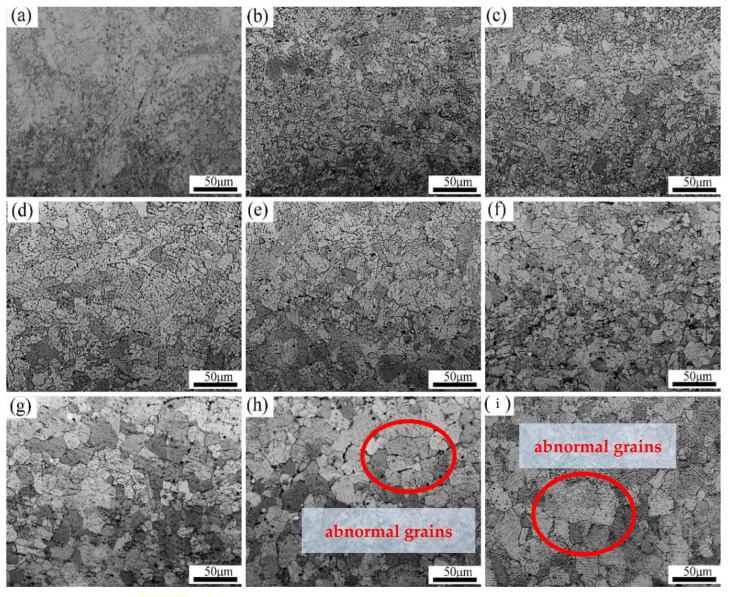
The metallography of 60% deformation with various annealing temperatures: (**a**) 460 °C, (**b**) 500 °C, (**c**) 540 °C, (**d**) 620 °C, (**e**) 660 °C, (**f**) 700 °C, (**g**) 740 °C, (**h**) 780 °C, and (**i**) 820 °C.

**Figure 11 materials-15-01370-f011:**
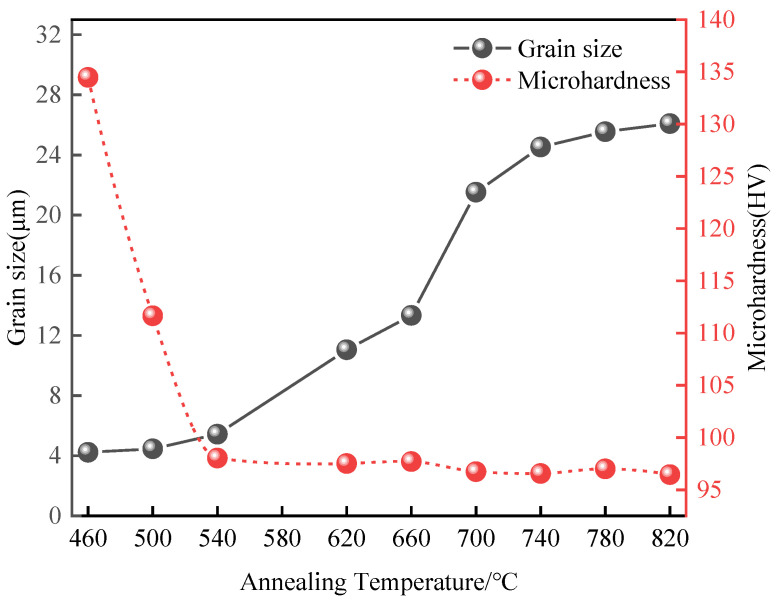
The relationship curve between annealing temperature with grain size and microhardness at 60% deformation.

**Table 1 materials-15-01370-t001:** Proportion of recrystallization area and deformation area of Er metal with various cold rolling deformation.

Deformation	Recrystallized (%)	Substructure (%)	Deformed (%)
30%	1.79	9.93	88.28
40%	3.44	0.35	96.21
50%	2.09	3.30	94.60
60%	6.60	8.84	85.09

**Table 2 materials-15-01370-t002:** The orientation index of crystal orientation under different deformation.

Deformation	$P\;(01\overset{-}{1}0)$	P (0001)	$P\;(\overset{-}{1}2\overset{-}{1}0)$	$P\;(11\overset{-}{1}0)$
30%	1.791	1.104	0.852	1.552
40%	1.600	1.220	0.945	1.617
50%	1.440	0.971	0.723	1.558
60%	1.184	1.120	0.681	1.771

**Table 3 materials-15-01370-t003:** Hardness and grain size of 60% deformation at different heat treatment temperatures.

Temperature (°C)	460	500	540	620	660	700	740	780	820
Microhardness (HV)	134.47	111.65	98.03	97.51	97.70	96.73	96.55	97.00	96.45
Grain size (µm)	4.23	4.46	5.43	11.05	13.33	21.52	24.53	25.55	26.09

## Data Availability

The data that supports the results of this study are available upon request from the authors.

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
