# Peer review of "Effect of Cold Rolling and Annealing on the Microstructure and Texture of Erbium Metal"

_materials, 2022, doi:10.3390/ma15041370_

Round 1
Reviewer 1 Report
I regret to inform you that the manuscript can not be accepted in the present form. It seems like the authors just ignored the data they have and made a story out of no where. Some comments are mentioned below;
1) The manuscript is not well managed and there are many types and serious flaws.
2) Abstract is not well written.
3) The authors mentioned the some grains are of 200 micron but the grain size distribution does not show any peak.
4) Section 3.2 is not well managed.
5) What is the criteria to select the recrystallized grains?
6) Fig 6 does not show ant twinning at any deformation conditions and the description of this fig is totally different form the Fig.
7) The other important discussion point of the paper is about the orientation/ texture. Why don't authors provide the pole figures of all the conditions?
Reviewer 2 Report
Reviewer’s comments
- The abstract section is quite good. However, it should be included more information about the next methods, concepts, results and conclusions on the novelty of the method.
- In the introduction section, the authors should state the motivation of the work related to your present study. The end of the introduction should be precise and concise given the research gap as well as the scope and limitations.
- In the experimental section, the authors should precisely give the details and phenomenon more clearly. It is too short.
- The figure should be clear and add details in the picture.
- In conclusion, the section should be shortened with new knowledge, technology, analysis, scope and limitation, which are concise with the journal scope in the body of the manuscript.

Reviewer 3 Report
- The main question of the paper consists in the materials science study of the specific metal after the certain mechanical-thermal treatment, which is often used for the usual structural metals and alloys. It distinguishes this research.
- This research opens a possibility to improve the functional properties of the metal on the base of material science approach.
- Just the used material science methodology allows to reach the main goal - improve the properties.
- The conclusions are consistent with the evidence and arguments
presented and they address the main question posed. - The Figure with a fine grain size after rolling enrich a great size for as-received state to point the rolling effect
Reviewer 4 Report
Dear Authors,
I have read your paper: "Effect of cold rolling and annealing on the microstructure and texture of erbium metal".
It fulfills the aims and scope of Materialsjournal. Presented investigations are interesting. My suggestions and comments are listed below.
General remarks:
- Please check the style of your paper. In some areas, it is different than template - e.g., in the abstract, there are different leading values (e.g., line 18-19 vs. 19-20).
- You have presented 32 references. However, only 5 of them have been published since 2020, one in 2021. I propose to add some of newest references. It will incease the further citations of your interesting work.
Introduction:
- In my opinion, this section is quite short and presented mostly very general information. Please describe the usage of investigated material - marh the most important applications. Moreover, please describe more deeply the problems with metal forming of your material.
Experimental:
- Lines 64-67 - why these temperatures vere selected? Please describe in the text.
- You made micrscopic tests and hardness measurements. These tests are widely described in relevant standards. Have you used requirements from any standards during tests? If yes, please mark in the text relevant standard numbers. If not, please describe, why the standards were not used in your investigations.
- Please speciffy, which Vickers scale were used (HV5, HV10, etc...).
Results and Discussion:
- The results are interesting. However, some misslacks appeared in your work.
- Some values are impossible to read from your figures (Fig. 8 and 10). Please show values near plots or in relevant tables.
- The discussion should be extended. Please compare your results with results from other papers. It will allow to mark the most important advantages from your investigations. Whan new has been found? Which problems could be eliminated by using your methodology? Please discuss in the text.
- Please support equations by relevant references.
Conclusions:
- This section is clear.
Round 2
Reviewer 1 Report
It is regret to inform that the revision is not significant enough to make this manuscript acceptible.
Reviewer 2 Report
The quality of paper is very much improved. The authors have revised according to the reviewer comments. I think that this paper is possible to meet the publication in the scientific journal.
Best regards,